# The Use of Mobile Applications for Sustainable Development of SMEs in the Context of Industry 4.0

**Angelina Iakovets [1,\*], Michal Balog [2] and Kamil Židek [1]**

[1] Department of Industrial Engineering and Informatics, Faculty of Manufacturing Technologies with a Seat in Presov, Technical University of Kosice, Bayerova 1, 08001 Presov, Slovakia

[2] Department of Logistics and Transport, Faculty of Mining, Ecology, Process Control and Geotechnologies, Technical University of Kosice, Komenskeho 19, 04001 Kosice, Slovakia

\* Correspondence: angelina.iakovets@tuke.sk

**Abstract:** Sustainable development of technology in manufacturing motivates entrepreneurs to increasingly introduce technical innovations into their production process. In times of technical progress, the selection and support of qualified personnel are especially important. The process of introducing new technologies or innovations in an enterprise is closely related to the personnel involved in this process. Sustainable development of the enterprise is possible, subject to the sequence of processes, namely, ensuring the adaptation of personnel and collecting feedback. A mobile application can be innovative for a particular enterprise and ensure sustainable development and adaptation to Industry 4.0 technologies. Given the pressure of technology and the environment on small- and medium-sized manufacturing enterprises, it was proposed to conduct a practical study in the conditions of a typical enterprise of this category. To explore the attractiveness of mobile applications as a tool for enterprise sustainable development, an application with basic features that should help provide a framework for integrating Industry 4.0 technologies into the manufacturing process was proposed. In the course of the study, a cycle of innovations and a set of evaluation methods for some of them were proposed. With the help of the proposed assessment methods, as well as the practical use of the mobile application, it became clear that the proposed solution can create a positive effect. The success of this kind of innovation and the further sustainable development of the enterprise is possible if the gradual adaptation of employees will be ensured; it will enable further innovation.

**Keywords:** mobile application; Industry 4.0; sustainability; management; manufacturing

## 1. Introduction

The economic growth of a country largely depends on the ability of this particular country to support the development of entrepreneurship. Small- and medium-sized enterprises (SMEs) play a key role in the development of entrepreneurship. There is a special demand for individual entrepreneurial activity in the European Union, where 37% of the population wants to engage in entrepreneurial activity [1].

SMEs contribute 40% to the GDP in countries with emerging economies [2]. The income received by SMEs and research has shown the importance of SMEs in the national economy of each country. SMEs occupy the biggest part of firms in most countries, especially in countries with developing economies, (about 80–95%) [1,2].

According to the World Trade Organization, SMEs represent over 90% of the business population, 60–70% of employment, and 55% of GDP in developed economies. As the world economy faces prevailing challenges, governments increasingly start to turn to SMEs as a significant element of sustainable and inclusive economic growth, namely for economic growth, poverty reduction, innovation, and job creation [3].

Modern society is undergoing a profound transformation under the influence of new technologies, and the manufacturing sector has undergone special changes. It is especially

difficult for small and medium-sized enterprises to maintain the required level of competitiveness as they need to pay special attention to modernization, the development of new technologies, products, and services, reduce the amount of waste, reduce resource consumption, and also invest in personnel with the right skills [4]. The introduction of the latest technologies can be a tool for solving the problem of SME competitiveness. Replacing paper carriers with digital ones opens up opportunities to ensure the continued operation of enterprises even during a pandemic, as well as accelerate the exchange of information within the enterprise and outside it. The most popular solution for manufacturing enterprises is the implementation of technologies of Industry 4.0 [5], which implies the use of wireless mobile technologies and the digitalization of processes. Mobile technologies can be represented by mobile devices that make possible the mobility of employees and automate the processes of obtaining and processing data by mobile applications. New technology implementation requires vertical and horizontal integration, which includes a lot of activities and most importantly time.

The widespread adoption of Industry 4.0 devices and technologies has increased the demand for technological innovations not only by manufacturers but to a greater extent by consumers. In the modern economy, the consumer forms the demand and characteristics of new products, and therefore enterprises try to organize their production to be able to flexibly respond to the changing consumers' needs. Features of the modern market can be summarized as:

- a consumer-oriented market,
- a highly competitive market,
- a market for a wide range of goods with a life cycle but a high level of technology,
- a market where production is characterized by full utilization of production capacities and use of low consumption technologies,
- a market characterized by a high level of service [6].

Thus, the main obstacles that need to be overcome by small- and medium-sized enterprises to take a competitive position in the market are the following:

- preparation and provision benefits of new technologies to employees,
- preparing the basis for the successful implementation of innovations,
- selection and preparation of the courses for personnel training programs,
- design a high-quality organizational structure of the enterprise to ensure the quality of services and products,
- effective planning.

Such criteria Chang et al. [7] define and evaluate by technology TOE (technology–organization–environment), multiple-criteria decision-making approach (MCDM), and influence network relations digraph (INRD) with accordance to DEMATEL analytic network process [7]. The research of Shiaw Tong Ha et al. [8] showed that three of four knowledge management (KM) dimensions, knowledge acquisition, knowledge conversion, and knowledge protection are positively related to enterprise performance. Summarizing the research of specialists in this field [5–7,9], it can be concluded that modernization according to Industry 4.0 depends on TOE and requires MCDM analysis to ensure a positive result from the implementation of the components of Industry 4.0.

E-business and information technology (IT) are critical components of improving logistics functions, Kamariotou et al. [9] confirm in their study, where they also note the importance of creating a high-quality enterprise information system. The importance of high-quality organization of enterprise information systems is due to its main task—continuous data exchange [9].

Investing large amounts of energy and money in major software updates and technological changes does not always bring the expected results. The automation process by software can be carried out in stages by small programs in the form of mobile applications. Although modern enterprises are increasingly replacing manual labor with automated,

sustainable development of HR management and support of human resources has become especially important.

Earlier, Jerry S.F. Lee [10] wrote about the importance of technology adoption. The importance of adaptation is very acute, even in fully automated production processes, because it still involves people, in one or another way. Moreover, it should be highlighted that the more high-tech process, the higher the requirements for the employee of the enterprise. The main factors that affect the process of adaptation to new technologies mainly associated with the human factor can be seen in Figure 1 [10].

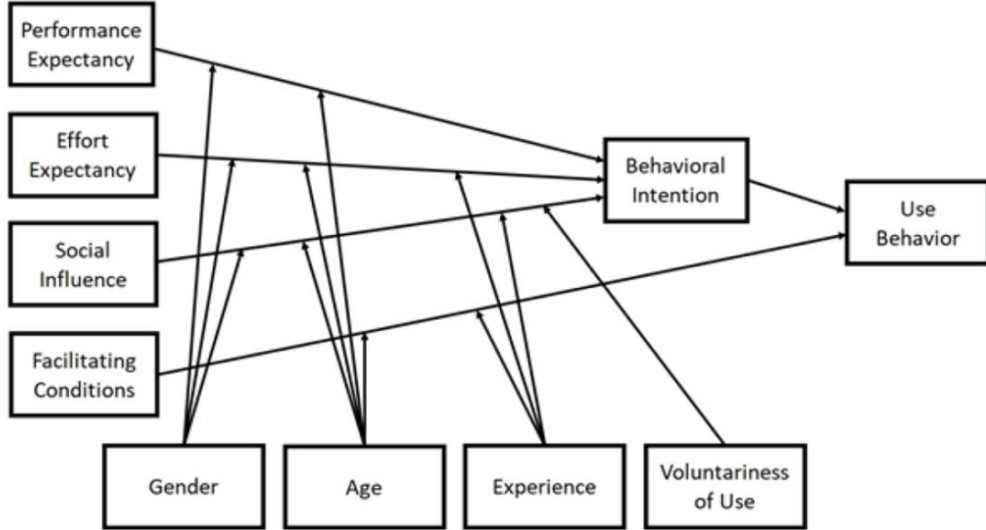

**Figure 1.** Theory of acceptance and use of technology [10].

Increasing requirements for employees are always associated with stress, which directly affects the quality of the enterprise [11]. The level and magnitude of stress depend not only on the technologies themselves but also on many additional factors, such as age, gender, experience, and voluntariness of use (Figure 1). Human resource management should be organized so that employees will be interested in developing not only their skills but also in the stable development of the entire enterprise [12]. Enterprises can partially relieve pressure on staff using covert data collection in the mobile application, which in turn will provide indirect feedback from employees. Data about quality of work, operating time and downtime, and equipment breakdowns can be quite easily collected by special sensors for automated reporting realized in the mobile applications [5].

The collection and timely processing of personnel data is especially important in the process of introducing new technologies to the enterprise. The work of employees is a powerful and significant potential of the company in terms of economic indicators, especially in SMEs [13].

Thanks to automation and the presence of a unified information base, all actions are carried out quickly, and most importantly, calculations, statistics, and analytics are always at hand. These conditions can be met by using mobile applications since their feature is the ability to integrate into enterprise information systems. Mobile devices can enable mobility of employees during the work day and help to save time on studying materials, reporting, analyzing and tracking processes, and also communication.

Automation of personnel management processes and training by mobile applications is also available and getting feedback by application is easier and faster than by traditional methods. The monetary drain of employee turnover costs manufacturers hundreds of thousands and sometimes millions of dollars annually [11,14].

A lot of enterprises face the challenge of finding new skilled workers. This is a multifaceted problem, as it relates to the rapid retirement rate of opulent employees, and a

shortage of qualified young workers due to the lack of graduate programs in intelligent manufacturing (robotics, automation, big data, analytics, and additive manufacturing) [5,14,15].

Manufacturing enterprises every day are faced with the problem of ensuring all these three processes because they are difficult to provide during work processes.

Implementation of technologies of Industrial Revolution (IR) 4.0 and HR development depends on:

- IR 4.0 training
- IR 4.0 company strategy
- IoT and cyber-physical systems awareness
- New skills aligned to IR 4.0
- Implementation of the latest emerging technology
- Training content satisfaction
- Skill and knowledge utilization
- Increased employee efficiency [16]

The introduction of IR technologies and staff training are closely interconnected, and therefore it is worth exploring the possibility of using the mobile application as an integrating tool of Industry 4.0 technologies.

## 2. Materials and Methods

### 2.1. Mobile Application Integration Stages

To ensure the equal and efficient implementation of new technologies, the so-called "vertical integration" should be used, which means ensuring uninterrupted communication and the flow of technological infrastructure used in all processes. A tool for "vertical integration" of technologies can be a mobile application that contains a knowledge base for employee training, and accountability, as well as collecting feedback from employees. The need for reporting and feedback is grounded on the need of enterprise management to have an overview of the current situation and to understand how well employees are trained, the level of study materials, and to get the overall picture of the correctness of the redistribution of duties among employees to achieve the expected labor productivity [17,18]. Feedback from users of the application as well as a group of experts can be collected using tests from IT as well as a general survey using the coefficient of importance of factors and points.

The introduction of something new always affects the mental mood of the team and that is why a priority task of managers is to design a learning process to encourage employees to develop themselves and learn something new. To ensure the comfortable adaptation of employees and sustainable management of human resources of the enterprise, it is worth organizing an adaptation plan. Enterprises under Industry 4.0 implementation have to go through a cycle of innovation every time (Figure 2) [19,20].

The proposed cycle can provide sustainable development of the manufacturing enterprise, due to implementing innovations oriented not only on automating but also on people involved in the manufacturing process.

The "Innovation" stage of the proposed study lies in using the mobile application as Industry 4.0 tool of vertical integration and a tool for adapting to new technologies.

Under implementing new solutions, it is important to explain (Figure 2, "Motivation") employees' essence of the changes, their importance for employees and the enterprise, and other changes associated with it, to ensure sustainable personnel management [19–21].

Using mobile applications is very similar to the M-learning process (mobile learning), where new technologies become a helpful tool for the adaptation process to new technologies [12]. The M-learning process is very simple due to its way of providing information and graphical illustration of the information [22].

The introduction of mobile IT applications in SMEs is associated with a combination of the technology–organization–environment (TOE) structure and is aimed at improving IT innovation and the implementation of Industry 4.0 technologies. That is why it is necessary to evaluate the effect of aspects of technology (technology readiness, technology security, and technology integration), or generation (financial commitment, organization readiness,

and top management support), and environmental criteria (regulatory support, competitive pressure, and environmental uncertainty) on Industry 4.0 technology application [7].

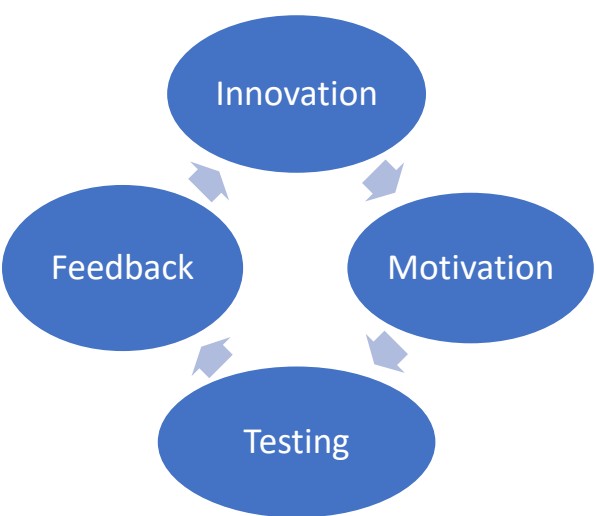

**Figure 2.** The enterprise's cycle of innovation.

The mobile application is a tool for implementing technologies of Industry 4.0 and is also a tool for M-learning [22–29]. According to this, there is also another aspect of evaluating such innovation—this is an assessment by the user or relevance for a particular group of users.

The enterprise under study has all features of manufacturing SMEs in the Slovak Republic:

- wide and non-constant range of products,
- training of personnel on the mentoring method using technical documentation and paper media,
- under the introduction of an increasing number of automated systems and mechanisms into production,
- quantity of employees is up to 250,
- two work shifts,
- main staff age group range is 25–70 years old,
- seasonal employees are hired with an increase in the number and volume of orders.

As part of the study, the main test group was the work shift of welders, since welding work had the greatest impact on the quality of the final product.

The stage of "Motivation" (Figure 2) should pick up arguments for employees. At this stage, it is important to explain that mobile applications, for users, have become commonplace for a long time and mobile work applications will not be something difficult. To ensure this stage, it is worth studying the characteristics of the personnel of manufacturing enterprises.

### 2.2. "Motivation" Stage

Since the proposed research was aimed at practical testing of the mobile application, as employee training and supporting tool in manufacturing enterprises, "Motivation" had to be supported not only by theory but also by the functionality of the application, which determined the main requirements for the mobile application sample:

- access to digital work instructions [29,30] and tests,
- two-level access to learning materials: for managers and operators,
- integrated data collection process about employees,
- automatic data exchange with the enterprise's database,
- feedback [31].

Such requirements were proposed to reduce requirements on users and reduce the engagement of humans in the data collection and processing process [29,30]. The emphasis on placing work instructions in the mobile application was justified by the fact that the company has mass production and the product range changes frequently. Changes in the range of products at the enterprise lead to constant training of employees and the creation of new work instructions.

All employees' activities and test results automatically will be put into the general table of personnel assessments on the server. The managers' interface includes some more options such as:

- quality reporting,
- adding tests,
- adding instructions,
- access to collected data from welders.

The structure of the mobile application was designed in accordance with the ERP program to reduce the time of data collection and proceeding.

ERP program is used by all employees; therefore, it was important to connect the data collected by the mobile application and the ERP program within the database. The connection was realized on the level of the server.

In theory and practice, a quantitative indicator of labor productivity is calculated with the use of strict values [32,33]. To ensure soft adaptation to new technologies, a combination of classical methods and data collected from the application, namely the start and end times of work shifts, was proposed. Such application features were supposed to reduce the amount of paper documentation and data collection time. Time losses, production defects, and production downtime—all these factors directly affect the profit of the enterprise and its place in the market [34]. Based on this information, it follows that there is an important factor in providing sustainable HR management and adaptation [35] to new technologies. The mobile application in this way should be tested on the biggest part of employees, welders, because they create product quality in the enterprise. Managers provide support to this group of employees and create "motivation", which is why it was decided to test the application in two steps: on managers and welders.

As "motivation" pros for managers, the following were selected:

- Corporate E-learning or M-learning [36] takes from 40% to 60% less time than the traditional learning process [25],
- 67% of people use mobile devices for access to E-learning courses and all active mobile phone users are 43 percent more productive than non-mobile devices users [34,35],
- Effectiveness of M-learning is proven by research by Globe News Wire, where it was written that in 2015, the mobile learning market was worth just 7.98 billion US dollars but by the end of 2019, that number had risen to 27.32 billion US dollars which proves that this market is promising. During the COVID-19 pandemic, there has been a steady worldwide increase in mobile users. Experts predict that the mobile and E-learning market will rise and the sector will maintain a Compound Annual Growth Rate (CAGR) of 36.3 percent down to 2025 [29,30,36],
- The growth of financial investments in M-education indicates that this method corresponds to modern trends in the technology market and makes possible a quick and affordable data exchange, which is especially important in manufacturing [5],
- Market research of employee engagement software has shown that the top 10 applications [37] and all these programs have some features that are very helpful in manufacturing enterprises such as:

  - customizable features,
  - easy to use,
  - feedback,
  - real-time score [37].

- The price of such applications ranges from 4 US dollars per month for one user to 20 US dollars [37]. As the proposed research was aimed at studying SMEs, it is reasonable to calculate the possible costs of implementing such an application. It is reasonable to take for calculating the maximum number of employees for a small enterprise (50 people) and the maximum number of employees for a medium enterprise (250 people) [32]. Paying 200 or 5000 US dollars per month for an application that is not tailored to the needs of a particular manufacturing enterprise is not a reasonable investment [35] which is why important to realize the research on the feasibility of a such decision.

The next stage was to determine the attractiveness of practical use of the mobile application for both groups of employees. The age group of workers under study and regular Slovak manufacturing enterprises is 25–70 years old, but most employees (about 90%) are in the age group of 25–55 years old. It is also necessary to study the group of the seasonal labor force, which mainly consists of people waiting for work in the labor market [38].

The biggest part of unemployed people in the Slovak republic is in the age group of 25–54 years.

According to information from Figures 3 and 4 and the information below, it can be said that the studied group of people consists of active smartphone users. Moreover, based on the above theoretical information, it is possible to hypothesize that the mobile application might be easy for using for the selected group of people.

| | Age groups | | | | | |
|---|---|---|---|---|---|---|
| | less than 25 years | | 25-54 years | | 55 years and more | |
| People pending employment | Less than 20 year | 5,024 | 25-29 years | 18,912 | 55-59 years | 21,042 |
| | | | 30-34 years | 19,678 | | |
| | | | 35-39 years | 20,555 | | |
| | 20-24 years | 16,389 | 40-44 years | 20,626 | 60 years and more | 10,646 |
| | | | 45-49 years | 19,131 | | |
| | | | 50-54 years | 18,864 | | |
| Total amount | 214,113 | | 117,766 | | 31,688 | |

**Figure 3.** Structure of unemployed people of Slovak Republic in 2020 [38].

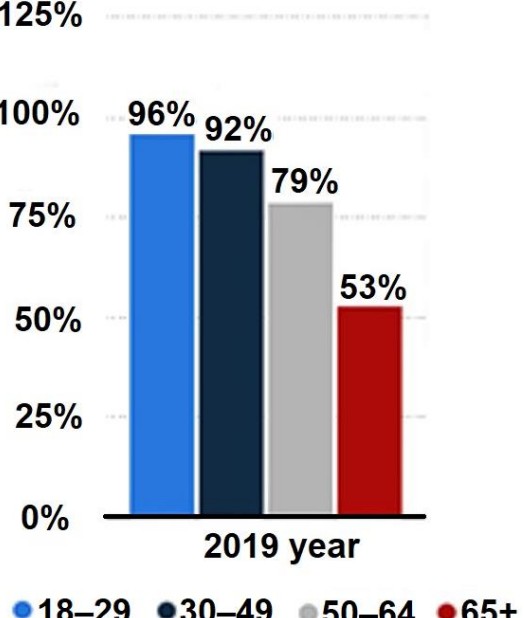

**Figure 4.** Smartphone users by age groups [34].

Management of the enterprise accepted the advantages of the proposed method of adaptation mentioned above and the mobile application went into the "Testing" stage (Figure 2) by the groups of users.

## 3. Results

*Background of "Testing" and "Feedback" Stages*

The proposed study is primarily aimed at evaluating the effectiveness of the mobile applications as a tool for adapting to new technologies in manufacturing SMEs. The production process in SMEs is based on workers and production managers and both are responsible for product and staff quality, which is why it was important to get feedback from both groups of users.

The first stage of "Testing" was to give the mobile application group of specialists (top management and stakeholders of the enterprise) and then was realized "Feedback" collecting process by evaluating its efficiency. This group was not chosen by chance, since in DEMATEL, MCDM, and INRD [7] theories and other assessment methods, these people are an important component of decision-making on automation.

The assessment was carried out using a questionnaire, where each criterion was assigned by a weighting factor as well as a scale of grades from 0 to 5, where "5" is the maximum value and 1 is the minimum one. To simplify the evaluation process, the original teaching method was called "mentoring" and the proposed was called "M-learning".

The consistency of expert opinions on each indicator is assessed by scattering estimates around the mean (variances) using Fisher's method.

The average score for each indicator "x" was calculated by the formula:

$$\overline{x} = \frac{\sum_{i=1}^{m} x_i}{m} \tag{1}$$

where $x_i$—mark is given by the i-th expert on this indicator;

*m*—number of experts.

There were 4 experts (project manager, welder's team leader, and two specialists from the university) and 5 criteria (Table 1).

**Table 1.** F-test of expert assessment of training.

| Criterion: | Value | Mentoring | Mentoring (X) | M-Learning | M-Learning (x) | $S_1^2$ | $S_2^2$ | F |
|---|---|---|---|---|---|---|---|---|
| The convenience of perception and understanding | 0.2 | 4 | 0.8 | 4 | 0.8 | 0 | 0.01 | 0 |
| | | 4 | 0.8 | 4 | 0.8 | | | |
| | | 4 | 0.8 | 5 | 1 | | | |
| | | 4 | 0.8 | 4 | 0.8 | | | |
| | Average | 4 | 0.8 | 4.25 | 0.85 | | | |
| Easiness of study | 0.2 | 5 | 1 | 4 | 0.8 | 0.01 | 0.013 | 1.33 |
| | | 5 | 1 | 3 | 0.6 | | | |
| | | 5 | 1 | 3 | 0.6 | | | |
| | | 4 | 0.8 | 4 | 0.8 | | | |
| | Average | 4.75 | 0.95 | 3.5 | 0.7 | | | |
| Need for additional advice | 0.2 | 1 | 0.2 | 2 | 0.4 | 0.03 | 0.013 | 2 |
| | | 0 | 0 | 1 | 0.2 | | | |
| | | 2 | 0.4 | 1 | 0.2 | | | |
| | | 1 | 0.2 | 2 | 0.4 | | | |
| | Average | 1 | 0.2 | 1.5 | 0.3 | | | |
| The convenience of displaying information | 0.1 | 4 | 0.4 | 5 | 0.5 | 0.0025 | 0.0025 | 1 |
| | | 4 | 0.4 | 4 | 0.4 | | | |
| | | 5 | 0.5 | 4 | 0.4 | | | |
| | | 4 | 0.4 | 4 | 0.4 | | | |
| | Average | 4.25 | 0.425 | 4.25 | 0.425 | | | |
| The quality of staff training | 0.3 | 4 | 1.2 | 5 | 1.5 | 0.0225 | 0 | 0 |
| | | 3 | 0.9 | 5 | 1.5 | | | |
| | | 3 | 0.9 | 5 | 1.5 | | | |
| | | 3 | 0.9 | 5 | 1.5 | | | |
| | Average | 3.25 | 0.975 | 5 | 1.5 | | | |

The consistency of expert opinions is assessed by the scatter of xi around the mean. To do this, it is necessary to calculate the variance of the estimate and compare the significance of its difference from the minimum possible variance according to Fisher's criterion.

The variance is calculated by the formula:

$$S_1^2 = \frac{\sum_{i=1}^{m}(x_i - \bar{x})^2}{m - 1} \tag{2}$$

The variance for mentoring ($S_1^2$) was 0.08 and 0.05 for M-learning ($S_2^2$).

The next step was the F-Test: hypothesis tests for the variances of 2 normal distributions.

$$F = \frac{S_1^2}{S_2^2} \tag{3}$$

To assess the quality of the proposed teaching methodology, it was hypothesized that M-learning is equivalent in quality to E-learning. To confirm or refute the theory, calculations were made according to the above formulas for each criterion.

The average X for mentoring is 3.4 and for M-learning is 3.7 (Table 1.). The average score differs by only 0.3 which is not significant, given the assessment step of 1 point.

Provided that $(X \sim F(m, n)) : P(X \leq F_\propto(m, n)) = \propto$, where $\propto = 0.9$, $n = 3$, and $m = 4$ the hypothesis is confirmed, since $\Phi$ of each parameter is less than its tabular value (5.34).

Implementing training using mobile devices, the employees' level of skills in working with modern gadgets has to be considered, since not every employee can be an active user of mobile devices. The need for additional advice can be seen on the right side of the graph, line "3" (Figure 5) corresponding to "The quality of staff training" from Table 1.

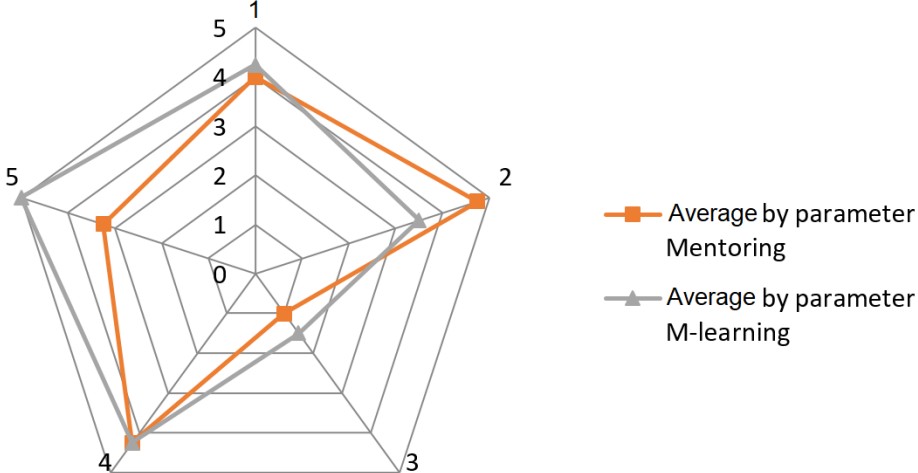

**Figure 5.** Comparison of mentoring and M-learning.

According to the experts evaluating (Figure 5), M-learning could replace hands-on learning (mentoring) because the mobile application is inherently multi-functional and able to expand the range of tasks to be performed.

After an expert evaluation of the suitability of the offered application and especially its content for the investigated company, the mobile application was handed over for practical testing to a group of 50 employees in the next part of "Testing".

After using the mobile application in the work process, selected welders evaluated the application by the Usability test [39], which is a standard test of the usability of the mobile application in practice. The system Usability Scale (SUS) is a base for the Usability test [39–41] and serves as a tool for getting feedback from users. This test is aimed at determining the level of usability, learnability, comprehensibility, and attractiveness of the target audience of product users [40].

SUS consists of 10 statements with 5 options for respondents (from strongly agree to strongly disagree) (Figure 6). The SUS evaluates a wide range of products and services, including hardware, software, mobile devices, websites, and applications [39]. The user's score for each statement is converted to a new number, added, and then multiplied by 2.5 to convert the original scores from 0–40 to 0–100. Although the scores are 0–100, they are not percentages and should only be considered in terms of their percentile ranking. According to the research, a SUS score of 68 and higher is considered to be above average, and a score that is less than 68 is considered to be below average.

The SUS questionnaire was completed by the group of target users. The average score of 50 SUS tests was 28 points. Then, this value was converted by multiplying by 2.5 and a score of 70 was obtained. Since 70 is more than 68, it means that the result is above average and satisfactory. The next step was to illustrate the conversion results (Figure 7).

PARTICIPANT NAME: ______________________________          DATE: ______________________________

## System Usability Scale

For each of the following statements, please mark one box that best describes your reactions to MSP today.

| | | Strongly disagree | | | | Strongly agree |
|---|---|---|---|---|---|---|
| 1. | I think that I would like to use MSP frequently. | 1 | 2 | 3 | 4 | 5 |
| 2. | I found MSP unnecessarily complex. | 1 | 2 | 3 | 4 | 5 |
| 3. | I thought MSP was easy to use. | 1 | 2 | 3 | 4 | 5 |
| 4. | I think that I would need the support of a technical person to be able to use MSP. | 1 | 2 | 3 | 4 | 5 |
| 5. | I found the various functions in MSP were well integrated. | 1 | 2 | 3 | 4 | 5 |
| 6. | I thought there was too much inconsistency in MSP. | 1 | 2 | 3 | 4 | 5 |
| 7. | I would imagine that most people would learn to use MSP very quickly. | 1 | 2 | 3 | 4 | 5 |
| 8. | I found MSP very cumbersome (awkward) to use. | 1 | 2 | 3 | 4 | 5 |
| 9. | I felt very confident using MSP. | 1 | 2 | 3 | 4 | 5 |
| 10. | I needed to learn a lot of things before I could get going with MSP. | 1 | 2 | 3 | 4 | 5 |

Comments (optional):

**Figure 6.** Template of SUS test of application.

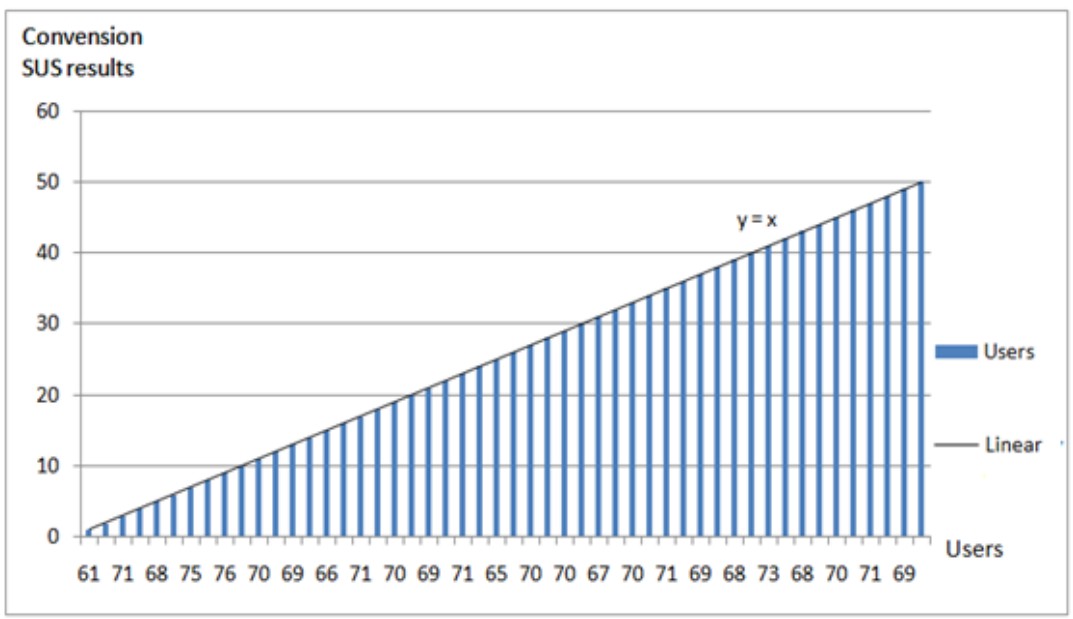

**Figure 7.** Conversion of SUS test.

According to the results of the SUS test, it can be stated that the experts and the target group of welders evaluate the application as effective. The next step was evaluation of nine g aspects (Figure 8), that can affect the process of the mobile application implementation.

| Aspects/Element | Descriptions |
|---|---|
| Technology aspect (*G1*) | |
| Technology readiness (*g1*) | Technology needs to be organized to make procedure changes, and latent sites must make changes for Industry 4.0 technology if benefits are to accrue. |
| Technology security (*g2*) | This refers to internet platforms being considered secure for conducting online transactions and exchanging data, including isolated data security and protection in using Industry 4.0 technology. |
| Technology integration (*g3*) | This refers to improving the responsiveness of information systems and decreasing incompatibility between legacy systems. |
| Organization aspect (*G2*) | |
| Financial commitment (*g4*) | This refers to the company that can offer the financial resources, and is especially committed to investing in employee training, software, hardware, system development, and system integration. |
| Organizational readiness (*g5*) | This refers to potential sites having to make decisions and businesses needing to be prepared to make business procedure changes for Industry 4.0 technology if benefits are to accrue. |
| Top management support (*g6*) | This refers to the top managers offering the support, and a promise to ensure a positive influence on this Industry 4.0 technology application procedure. |
| Environment aspect (*G3*) | |
| Competitive pressure (*g7*) | This refers to the adoption of Industry 4.0 technology, so that companies can benefit from more accurate data collection and a greater operational efficiency. |
| Regulatory support (*g8*) | This refers to elements of conception such as the policies of the government, which impact the diffusion of IT. |
| Environmental uncertainty (*g9*) | This refers to the environmental uncertainty, and how managers and entrepreneurs tend to act proactively according to well-informed conjectures about the strategic path ahead. |

**Figure 8.** Clarification of aspects [7].

Evaluation of mentioned aspects will show the strongest aspects and management will be able to provide sustainable modernization and development.

Mobile devices and mobile applications are components of Industry 4.0 and their implementation is associated with many factors that affect the conditions of their use, as well as the process of their implementation. Based on this, it was decided to turn to the research of Chang and colleagues and use elements for assessing the Industry 4.0 technology application effectiveness. This method is based on MCDM and DEMATEL combination.

For building the final graph of the effect of the provided implementation of technology on the manufacturing process using the expert method and evaluating of g factors affect [7], g is a set of factors related to the Industry 4.0 technology application within each aspect that is subsequently evaluated by 5 experts (stakeholders and top management) (Figure 8). The normalized average matrix of the evaluation is shown below (Table 2).

**Table 2.** Matrix of normalized average values of expert ratings.

| Average Matrix | g1 | g2 | g3 | g4 | g5 | g6 | g7 | g8 | g9 |
|---|---|---|---|---|---|---|---|---|---|
| g1 | 0 | 2 | 1.4 | 0.8 | 1.6 | 0.4 | 0.2 | 0.4 | 0.2 |
| g2 | 0.6 | 0 | 1.6 | 0.8 | 1.8 | 0.2 | 2 | 0.4 | 0.8 |
| g3 | 1.6 | 1.8 | 0 | 1.2 | 0.6 | 0.6 | 0.2 | 0.2 | 0.2 |
| g4 | 1 | 0.2 | 0.8 | 0 | 1.2 | 2.2 | 0.6 | 0.8 | 1.4 |
| g5 | 1.6 | 0.6 | 1.4 | 0.8 | 0 | 2.6 | 0.4 | 1.2 | 1 |
| g6 | 1.4 | 0 | 1.8 | 0.2 | 1.6 | 0 | 0.4 | 0.6 | 0.4 |
| g7 | 0.2 | 0.2 | 0.2 | 0.8 | 1.8 | 0.6 | 0 | 0.6 | 1.2 |
| g8 | 0.4 | 0.2 | 2.6 | 1.8 | 0.4 | 0.4 | 0 | 0 | 0.2 |
| g9 | 1.2 | 2.2 | 1.2 | 1 | 1.4 | 2.6 | 1.8 | 0.4 | 0 |

The final DEMATEL MCDM analysis can be represented in Table 3.

**Table 3.** Results of the DEMADEL MCDM analysis.

| | $R_i$ | $C_i$ | $R_i + C_i$ | $R_i - C_i$ | Identify |
|---|---|---|---|---|---|
| g1 | −1.18592 | −1.41569 | −2.60162 | 0.229772197 | Cause |
| g2 | −1.24655 | −1.41866 | −2.6652 | 0.172108986 | Cause |
| g3 | −1.13962 | −1.40314 | −2.54276 | 0.263520535 | Cause |
| g4 | −1.16896 | −1.01881 | −2.18778 | −0.150149811 | Effect |
| g5 | −1.17654 | −1.40442 | −2.58096 | 0.227883863 | Cause |
| g6 | −0.99864 | −1.0799 | −2.07853 | 0.081262193 | Cause |
| g7 | −1.06374 | −0.92671 | −1.99045 | −0.137021864 | Effect |
| g8 | −0.94888 | −0.85661 | −1.80548 | −0.092269376 | Effect |
| g9 | −1.4399 | −0.84479 | −2.28468 | −0.595106724 | Effect |

Table 3 represents four factors that can affect the final implementation of the proposed solution. g-factors that affect the implementation process are illustrated in Figure 9 by red dots.

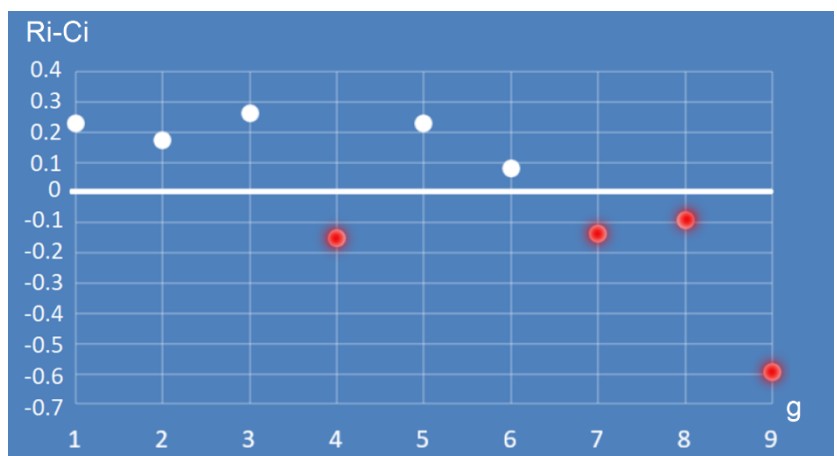

**Figure 9.** Graphical illustration of the DEMADEL MCDM analysis results.

The red dots are g4 (financial commitment), g7 (competitive pressure), g8 (regulatory support), and g9 (environmental uncertainty) which can be seen in Figure 9. According

to the graph, it can be seen that factor g9 has the greatest impact on the implementation process. This factor is integral during the implementation of the components of Industry 4.0, this is explained by the fact that technologies are developing very quickly and so is the software. DEMATEL MCDM method is based on the assessment of the influence of factors on one another and therefore, it is important to find not only the most influential factors but also to emphasize their relationship.

To draw up a diagram of the influence of factors on each other, it is worth referring to the Table 4.

**Table 4.** Total reversion matrix of DEMADEL MCDM analysis results.

| T Total Reversion Matrix | g1 | g2 | g3 | g4 | g5 | g6 | g7 | g8 | g9 |
|---|---|---|---|---|---|---|---|---|---|
| g1 | **−0.71202** | 0.12849 | −0.16953 | −0.08183 | 0.01537 | **−0.26235** | 0.02985 | −0.07672 | −0.05718 |
| g2 | **−0.34181** | −0.85330 | **−0.27959** | −0.15632 | 0.01736 | −0.04786 | **0.30742** | −0.02816 | 0.13572 |
| g3 | −0.08796 | 0.10679 | −0.79430 | −0.02976 | −0.08048 | −0.19448 | 0.08999 | −0.13870 | −0.01072 |
| g4 | 0.02394 | −0.21387 | −0.07675 | **−0.60679** | −0.15609 | 0.15783 | **−0.24592** | −0,02674 | −0.02458 |
| g5 | 0.08845 | −0.01721 | 0.06542 | −0.05823 | **−0.75647** | −0.05175 | **−0.28315** | 0,01490 | −0.17849 |
| g6 | 0.15497 | 0.04506 | 0.15685 | −0.09297 | −0.01575 | −0.77666 | **−0.20290** | −0.05466 | **−0.21257** |
| g7 | −0.15796 | −0.36523 | −0.19757 | −0.06894 | 0.00859 | 0.14514 | **−0.60098** | 0.07296 | 0.10026 |
| g8 | −0.09838 | −0.05444 | **0.28359** | **0.43427** | −0.39739 | −0.23115 | **−0.27824** | **−0.48739** | −0.11975 |
| g9 | **−0.28493** | −0.19494 | −0.39125 | **−0.35824** | −0.03955 | 0.18137 | **0.25721** | −0.13209 | **−0.47748** |
| Ci | **−1.41569** | **−1.41866** | **−1.40314** | **−1.01881** | **−1.40442** | **−1.0799** | **−0.92671** | **−0.85661** | **−0.84479** |
| Theresold (alpha) value | 0.20262 | | | | | | | | |

Figure 10 shows that not only g9 factor is important (Figure 9) but g8 is important too. This can be explained by the fact that with the development of technology, the area of collected and processed data is constantly expanding, which entails changes in legislation, as well as the facts of the adoption of legal restrictions on the use of powerful computers, which have increased power consumption [42], etc. Laws affect the very development of technology and technology, as well as the emergence of the Industry 5.0 trend. In the case of the studied enterprise and SMEs in general, it can be said that mobile applications and devices are the keys to the implementation of Industry 4.0, but one should not forget to take into account external factors of influence during vertical and horizontal integration.

According to the results of the study, it can be stated that mobile applications can be used as a tool for adapting to modern technologies and offered cycles (Figure 1) and methods of evaluation could be useful for other enterprises in the manufacturing branch. The proposed cycle of innovations will ensure sustainable development of the enterprise, since even highly automated manufacturing plants involve people in one way or another, and it is especially important to ensure that they adapt to the proposed changes.

Mobile applications, due to their wide functionality, can be used as a tool for vertical and horizontal integration of Industry 4.0 components into the manufacturing process.

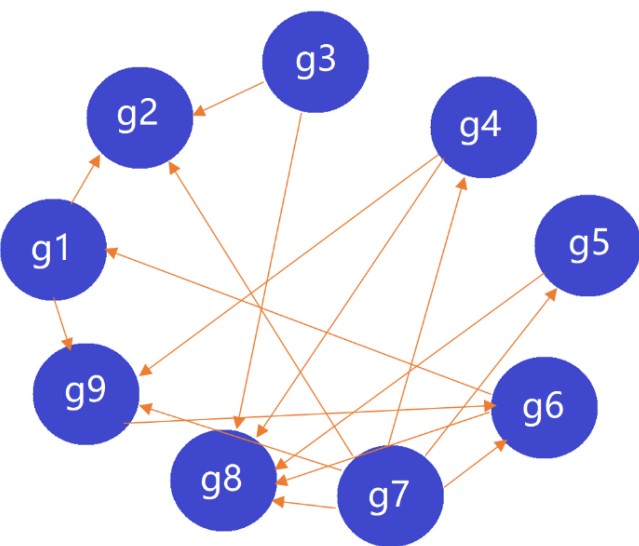

**Figure 10.** Graphic scheme diagram of the influence of factors on each other.

## 4. Conclusions

At the stage of developing the application content, the company's management took the idea positively and actively entered the stage of employee motivation. At the stage of application testing, it became obvious that the implementation of work instructions and basic functionality, such as registering an employee in the system, is just a small part of the functions that can be integrated into the application.

The introduced mobile application, only during the month of active use has shown a positive effect on the manufacturing process:

- the number of mistakes made by welders decreased by 18%,
- staff training time has been reduced from 1 h 30 min to 30 min,
- reduced production downtime associated with additional consultations on instructions from 1 h per work shift to 30 min,
- reduced the number of paper carriers of instructions,
- paperwork has been accelerated,
- data exchange become faster because the shift manager stopped spending 40 min of the shift collecting data from each workplace.

Workers noted that it was especially convenient that they were able to open necessary instructions and put them in front of them as a sample, according to which they made work and compared the quality of work performed. Visual control of the completed work is an integral part of the workflow of each employee, and the mobile application was useful for these purposes as well. The positive assessment of the application is especially emphasized by the results of the Usability test given in the section above.

Taking into account the fact that mobile application is primarily a program, it can be changed at any time according to the requirements of modernity.

A lot of manufacturing enterprises try to introduce the newest technologies in manufacturing processes, but they are every day faced with the problem of employee adaptation. The proposed study shows that mobile applications and mobile devices, as a component of Industry 4.0, can be accepted by employees of different ages, education, and professions. Feedback collected from specialists, management, and employees clearly showed the attractiveness of the proposed solution. Methods for analyzing the effectiveness of the mobile application used in the article can be used by other enterprises in the industry. It is worth emphasizing that to ensure the sustainable development of the enterprise, the proposed assessment methods should be applied according to the innovation cycle to ensure the adaptation of the personnel who will be involved in this.

Mobile devices are distinguished by their autonomy, and therefore, by expanding the functionality of the mobile application, it is possible to replace some devices for collecting and processing data in the enterprise, thereby reducing electricity costs and increasing the mobility of employees in the enterprise. The price of such a device will not be high because such a mobile application does not require high technical requirements, which is why the proposed mobile application can be introduced by many SMEs.

**Author Contributions:** Conceptualization, A.I. and M.B.; methodology, A.I. and K.Ž.; validation, K.Ž. and A.I.; formal analysis, M.B.; investigation, A.I.; resources, K.Ž.; data curation, M.B.; writing—original draft preparation, A.I.; writing—review and editing, A.I.; visualization, A.I.; supervision, M.B.; project administration, K.Ž.; funding acquisition, K.Ž. All authors have read and agreed to the published version of the manuscript.

**Funding:** This work was supported by the Slovak Research and Development Agency under contract No. APVV-19-0590, and by the projects VEGA 1/0700/20, KEGA 055TUKE-4/2020 granted by the Ministry of Education, Science, Research, and Sport of the Slovak Republic.

**Data Availability Statement:** Related works can be found here: https://doi.org/10.1007/978-3-030-90462-3_4 and DOI:10.1007/978-3-030-67241-6_6.

**Acknowledgments:** This work was supported by the Slovak Research and Development Agency under contract No. APVV-19-0590, and by the projects VEGA 1/0700/20, KEGA 055TUKE-4/2020 granted by the Ministry of Education, Science, Research, and Sport of the Slovak Republic.

**Conflicts of Interest:** The authors declare no conflict of interest.

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
