# Peer review of "The Use of Mobile Applications for Sustainable Development of SMEs in the Context of Industry 4.0"

_applsci, doi:10.3390/app13010429_

Round 1
Reviewer 1 Report
The article has important potential for publication. Some aspects need to be better organised to facilitate the reader's understanding. I will punctuate the points for improvement in order to facilitate the improvement work by the authors:
i. General revision by some native English proofreader. Sometimes it is difficult to understand the meaning of sentences. It is noticeable the truncated sentences;
ii) I suggest to avoid subsections in the introduction and to make clear the problematic and the research opportunities. Besides, it should clearly explain the objective of the research and, mainly, the contributions that the paper generates in terms of literature. These aspects are lost in the subsections that were inserted.
iii) There are aspects of the methodological procedures that are in the exposition of the results. Such aspects need to be separated. All methodological decisions, decision criteria, data collection and analysis procedures need to be in the Materials and Methods section.
iv) The results section needs to focus specifically on the analyses of the data obtained and its relevance to the objective of the study. It becomes difficult to comprehend the set of various results in terms of supporting the objective of the paper. The integration of the various results to support the objective needs to be clearly defined.
v) Conclusions do not confront the empirical results of the study with the literature analysed. This needs to be profoundly improved.
Again, the paper has good potential but needs substantive improvement for advancement.
Author Response
I am very grateful to You for Your valuable advice and for the positive assessment of the proposed research potential. Your remarks helped streamline the text of the proposed article. Thank You.

Reviewer 2 Report
Let me start with an important note: I think that the Title and the Abstract do not represent the paper content at all: in fact, by reading the Title and the Abstract a reader is expecting the description of at least one Mobile Application in terms of project, development and implementation, but the paper cites only its use, not even at the interface level.
About the structure of the paper, I suggest to write a real Introduction where the goals of the paper will be clearly introduced, together with the structure of the paper itself (such as; in section 2 we will report …, and so on). This kind of Introduction, in general, has a length between half page to one and half pages and, again in general, does not present sub-sections!
Some more specific comments:
1) The title of sub-section 1.2 (line 62) is confusing: perhaps it can be simplified in “The concept of Industry 4.0 in SMEs”.
2) At line 69, perhaps it is better to add the article “a” at the beginning.
3) At line 77, it is not clear the presentation at which people is addressed.
4) At line 85, the acronym in the parentheses is written in a wrong way: it is INRD and not INDR!
5) The phrase that starts at line 89 is unreadable!
6) At line 105, perhaps the verb “has” has to be substituted by “have”.
7) Again, the phrase at line 109-110 is unreadable!
8) At line 130, it is used the acronym MSP, without explaining its meaning; on google I found at least 3 meanings: Managed Service Provider, Managed Service Program, Minimum Support Price, what is the right one? Another one?
9) For a paper which title is “Mobile applications …” the fact that the first time the authors uses this term (apart from the Abstract) is on the line 133 at page 4 is a bit strange.
10) At line 140 which is the subject of “will help”?
11) Again, the phrases at line 157-161 are unreadable!
12) Please control if is correct at line 202 “should to go”.
13) In Figure 2, the word Innovation is written in a wrong way (Inovation!).
14) At line 209 is introduced the term M-learning (with the m in uppercase), but at lines 211 and 222 is reported with the m in lowercase!
15) At line 221, perhaps it is better to add the article “The” at the beginning.
16) Again, the phrase at line 222-223 is unreadable!
17) At line 242, the verb “introduces” it is not clear at what subject is referred.
18) At line 251, is “this position” correct?
19) At line 257, the authors say “According to the algorithm in Figure 1, …” but perhaps the figure to be cited is the 2, since Figure 1 seems does not represent an algorithm. The same citation problem seems present at line 307. As more wide problem, the authors have to consider the introduction, in the right places, of the reference to each element of figure 2 and not only for the testing stage.
20) At line 268-269, the authors write that the experts predict by the end of 2020 something, but we are at the end of 2022!
21) In the caption of figure 4, perhaps it is more clear use “age groups” (as used in the previous figure) instead of “age structure”.
22) At line 301, perhaps the figures that have to be referred are the 3 and 4, and not the 2 and 3!
23) At line 323, the authors report that data are collected by welders, but the reader is on the page 8 and it is the first time that these workers are mentioned!
24) The reference 39 is not cited in the paper.
25) Again at line 327, perhaps before “mobile application” the article “the” is missing.
26) At line 347, the figure 5 has to be referred in the parentheses.
27) At line 403, the authors say “The need for … can be seen on the right side of the graph (see Figure 7)” but this is not very evident. With regard to the figure 7, I thing that the term used in the caption is written in a wrong way (EVARAGE?)
28) Again, the phrase at line 415-417 is unreadable!
29) Why in the Figure 8, taken by [23], the last line corresponding to g9 is not reported?
30) Even if for me is not clear because in [23] had been used sometimes uppercase G and sometimes lowercase g, I think that the type of letter is important, since there is a G1 and a g1 in the table, also reported in your paper. Therefore, I think that the reader should unable to understand the meaning of table 2 and 3 that report only G in uppercase, from G1 to G9, considering that in [23] exists only G1, G2 e G3 in uppercase!
31) At line 426 which is the subject of “can be seen”?
32) At line 443, perhaps the figure to be cited is the 10, and not the 11.
33) At line 455, perhaps the correct term has to be “mobile applications” and not “mobile application”
34) At line 458 which is the subject of “can be said”?
35) At line 467, perhaps it is better to add the article “The” at the beginning.
Author Response
Thank You for Your professional review. Your valuable advice helped improve the article and logically organize the text of the article.

Round 2
Reviewer 2 Report
I highly appreciate that the authors have tried to answer to each of my points, but I have to underline that, in some cases, I have not found the indicated corrections in the paper, perhaps because the questions I have put should be not understood. To this regard, I only report the points of my previous review that result again not solved: 5, 6, 10, 26, 27 and 28.
As more general comments:
A) I note that the title has not changed: surely, the authors have added several phrases in the paper that try to explain the use of mobile applications, but I think that it is not again clear if the paper refers to a single mobile application or to a set of mobile applications. In fact, in some points of the paper the authors wrote “a mobile application”, in some points “the mobile application” and in others “mobile applications”, then this aspect is not clear. In any case, I think that the title has to be changed for example in “The use of mobile applications for sustainable development of SMEs in the context of Industry 4.0” to make more evident that the paper does not present the design of a specific application or of a set of applications, but only its/them use!
B) The actual Introduction, even if the authors eliminate its sub-sections, for me does not represent an Introduction to a paper since its content is more adequate to be put in a section of the paper with a different name. In fact, an Introduction has to be more concise and has to finish with the structure of the paper. For your convenience, I report that You can found in the Authors Guidelines (https://www.mdpi.com/journal/applsci/instructions#preparation) in particular in the Research Manuscript Sections, where I underline the more significant words - Introduction: The introduction should briefly place the study in a broad context and highlight why it is important. It should define the purpose of the work and its significance, including specific hypotheses being tested. The current state of the research field should be reviewed carefully and key publications cited. Please highlight controversial and diverging hypotheses when necessary. Finally, briefly mention the main aim of the work and highlight the main conclusions. Keep the introduction comprehensible to scientists working outside the topic of the paper.
C) Please carefully reread the paper to discover the phrases that do not have any subject! For example (referring to the point 5 not solved as I say before), “If summarize the latest research, can be said that modernization always depends on TOE and requires MCDM analysis for better planning and organization of modernization processes, to ensure a positive outcome” or “Should be noticed that any increase in productivity or efficiency due to new technology is impossible without a plan for onboarding employees and providing employee feedback.” only to make some examples.
Author Response
Point 1: I highly appreciate that the authors have tried to answer to each of my points, but I have to underline that, in some cases, I have not found the indicated corrections in the paper, perhaps because the questions I have put should be not understood. To this regard, I only report the points of my previous review that result again not solved: 5, 6, 10, 26, 27 and 28.
5) The phrase that starts at line 89 is unreadable!
6) At line 105, perhaps the verb “has” has to be substituted by “have”.
10) At line 140 which is the subject of “will help”?
26) At line 347, the figure 5 has to be referred in the parentheses.
27) At line 403, the authors say “The need for … can be seen on the right side of the graph (see Figure 7)” but this is not very evident. With regard to the figure 7, I thing that the term used in the caption is written in a wrong way (EVARAGE?)
28) Again, the phrase at line 415-417 is unreadable!
Response 1: Your comment is very valuable, thank You. I miss these points because the whole text was under changes and some of the lines change their number. Corrections:
5) Rewrited sentence can be seen on lines 90-93.
6) I wasn’t sure about “have” or “has”, that is why I decided to use a verb in past form (see line 107)
10) “will help” was changed, which can be seen on line 142
26) The number 5 already gets its parentheses, which can be seen on line 343. The same problem was with the sentence 379-380 line with 3. Thank You for reminding this problem, it helped me to find more mistakes.
27) The need for additional advice is evident due to the point on the chart. Word “evarage” was changed to the right form – “Average”, thank You for Your remark.
28) Mentioned lines were partially changed.
Point 2: I note that the title has not changed: surely, the authors have added several phrases in the paper that try to explain the use of mobile applications, but I think that it is not again clear if the paper refers to a single mobile application or to a set of mobile applications. In fact, in some points of the paper the authors wrote “a mobile application”, in some points “the mobile application” and in others “mobile applications”, then this aspect is not clear. In any case, I think that the title has to be changed for example in “The use of mobile applications for sustainable development of SMEs in the context of Industry 4.0” to make more evident that the paper does not present the design of a specific application or of a set of applications, but only its/them use!
Response 2: Thank You for the important remark, I fully accept Your recommendations. The article's name was changed according to Your remark.
Point 3: The actual Introduction, even if the authors eliminate its sub-sections, for me does not represent an Introduction to a paper since its content is more adequate to be put in a section of the paper with a different name. In fact, an Introduction has to be more concise and has to finish with the structure of the paper. For your convenience, I report that You can found in the Authors Guidelines (https://www.mdpi.com/journal/applsci/instructions#preparation) in particular in the Research Manuscript Sections, where I underline the more significant words - Introduction: The introduction should briefly place the study in a broad context and highlight why it is important. It should define the purpose of the work and its significance, including specific hypotheses being tested. The current state of the research field should be reviewed carefully and key publications cited. Please highlight controversial and diverging hypotheses when necessary. Finally, briefly mention the main aim of the work and highlight the main conclusions. Keep the introduction comprehensible to scientists working outside the topic of the paper.
Response 3: I have great respect for You and Your professional opinion. Your remarks have a place and I agree with them, but unfortunately, at the moment I did not have the opportunity to sit down and go through the text of the article in detail with my colleagues to correctly shorten the introduction of the article. The structure in the previous adjustment phase was subject to change, where the Cycle of Innovation became a support for streamlining the materials of practical research. If in Your professional opinion, the article is still far from the required level, I would be happy to ask you to give me time for corrections until December 5, if possible.
Point 4: Please carefully reread the paper to discover the phrases that do not have any subject! For example (referring to the point 5 not solved as I say before), “If summarize the latest research, can be said that modernization always depends on TOE and requires MCDM analysis for better planning and organization of modernization processes, to ensure a positive outcome” or “Should be noticed that any increase in productivity or efficiency due to new technology is impossible without a plan for onboarding employees and providing employee feedback.” only to make some examples.
Response 4: Thank You for Your professional and important remark. After discussion with co-authors of the proposed article, was decided to remove sentences on 102-103 lines.
Round 3
Reviewer 2 Report
I again appreciate the different interventions the authors made on the paper, but still there are some parts that need clarification before the paper can be acceptable for publication.
I try to list all of them, specifying the most important ones with a specific note:
1) IMPORTANT-The Abstract and the Introduction again do not clearly explain the goal/aim of the paper; it seems very strange that the authors, asked to rewrite the introduction, do not clearly state it. The reader should expect to find written, in particular in the Introduction, that the goal/aim of the paper is to demonstrate that using mobile applications in SMEs can lead to a set of advantages; moreover, the reader should expect to find a clear explanation that to demonstrate that thesis, ONE SPECIFIC MOBILE APPLICATION has been introduced in a specific enterprise to verify the impact of its use. In fact, in some points of the paper is not so clear if in the chosen enterprise only one mobile application has been used or several of them!
2) IMPORTANT-As I said in the first and in the second review, the introduction is too long and several parts of it have to be shifted in other parts of the paper or to be deleted. In any case, Introduction does not report the overall organization of the paper: it is common to find, at the end of the introduction of a technical paper, a part such as “In the first section, we analyze …; in the second section, we introduce …; etc.
3) IMPORTANT-Again there are some phrases that report the verb before the subject or that do not have any subject: these problems make difficult to understand the meaning of these phrases and then of the paper in total. Only to cite some of them: for the first problem “In the course of the study, was proposed a cycle of innovations and a set of evaluation methods for some of them.” That has to be rewriting as “In the course of the study, a cycle of innovations and a set of evaluation methods for some of them WERE proposed.” Or in alternative “Our study/research proposed a cycle of innovations and a set of evaluation methods for some of them.” For the second problem, “Also should be highlighted that the more high-tech process,… ” before “should” has to be put “IT”!
4) At line 81, the abbreviation KM is not clear since K is clearly “knowledge” but in the following there is “dimensions” that do not begin with M! In the same phrases, the word “dimensions” is repeated twice.
5) At line 103, the word “bellow” is written in a wrong way and, in any case, it is always better to refer to the number of figure that in this case is Figure 1!
6) At lines 108 and 109, the word “age” is repeated twice.
7) At line 109, the word “resourse” is written in a wrong way.
8) At line 123, the word “anable” is written in a wrong way.
9) Lines 184-187 perhaps contain a redundant explanation considering the following parts of the paper.
10) At lines 243 (as already I said in the first revision) the fact that the chosen enterprise (which characteristics are only introduced at line 207) has welders as workers has to be specified in the enterprise description!
11) At lines 255-257, the phrase “Based on this information, it follows that an important factor in providing sustainable HR management and adaptation [34] to new technologies.” Seems not completed!
12) At line 272, the acronym CAGR has to be explained!
13) At line 321, the term DEMATEL appears here for the first time: why is it not been introduced at lines 78-86 together with MCDM and INRD?
14) At line 347, what is X? Moreover, the results reported in the text, i.e. 3.4 for mentoring and -3.7 (with the minus?), where can be found in the table 1? And again, more in general, the meaning of table 1 is a bit obscure!
15) At line 372, please do not use the phrase “(see picture below)” but refer directly to the figure 6!
16) At line 388, how can the reader understand the meaning of the term “g9 aspects” since the explanation of figure 8, that contains these aspects, is after this part? In addition, it is not at all clear in the rest of the paper if the term “g9” refers to the single aspect named in this manner or refers to the set of the 9 aspects named with letter g followed by a number!
17) At line 405, which are the four factors that are reported in Table 3?
Author Response
I want to thank You for Your patience and hard work. Your professionalism was highly appreciated by me and my colleagues.
Thank You for taking the time to write such a professional review.
With regards.
